# Dynamic Transmission of *Staphylococcus Aureus* in the Intensive Care Unit

**DOI:** 10.3390/ijerph17062109

**Published:** 2020-03-22

**Authors:** Claire E. Adams, Stephanie J. Dancer

**Affiliations:** 1Dept. of Critical Care, Edinburgh Royal Infirmary, NHS Lothian, Edinburgh EH16 4SA, UK; claire.adams01@gmail.com; 2School of Applied Sciences, Edinburgh Napier University, Edinburgh EH11 4BN, UK; 3Dept. of Microbiology, Hairmyres Hospital, NHS Lanarkshire G75 8RG, UK

**Keywords:** *Staphylococcus aureus*, critical care, transmission, hand-touch site, ventilation

## Abstract

*Staphylococcus aureus* is an important bacterial pathogen. This study utilized known staphylococcal epidemiology to track *S. aureus* between patients, surfaces, staff hands and air in a ten-bed intensive care unit (ICU). Methods: Patients, air and surfaces were screened for total colony counts and *S. aureus* using dipslides, settle plates and an MAS-100 slit-sampler once a month for 10 months. Data were modelled against proposed standards for air and surfaces, and ICU-acquired staphylococcal infection. Whole-cell genomic typing (WGS) demonstrated possible transmission pathways between reservoirs. Results: Frequently touched sites were more likely to be contaminated (>12 cfu/cm^2^; *p* = 0.08). Overall, 235 of 500 (47%) sites failed the surface standard (≤2.5 cfu/cm^2^); 20 of 40 (50%) passive air samples failed the “Index of Microbial Air” standard (2 cfu/9 cm plate/h), and 15/40 (37.5%) air samples failed the air standard (<10 cfu/m^3^). Settle plate data were closer to surface counts than automated air data; the surface count most likely to reflect pass/fail rates for air was 5 cfu/cm^2^. Surface counts/bed were associated with staphylococcal infection rates (*p* = 0.012). Of 34 pairs of indistinguishable *S. aureus*, 20 (59%) showed autogenous transmission, with another four (12%) occurring between patients. Four (12%) pairs linked patients with hand-touch sites and six (18%) linked airborne *S. aureus*, staff hands and hand-touch sites. Conclusion: Most ICU-acquired *S. aureus* infection is autogenous, while staff hands and air were rarely implicated in onward transmission. Settle plates could potentially be used for routine environmental screening. ICU staphylococcal infection is best served by admission screening, systematic cleaning and hand hygiene.

## 1. Introduction

There is a limited evidence base for everything that we do in the name of infection prevention and control. This is because infection control itself is a very young science and has not yet amassed sufficient evidence to achieve universally agreed practices. One of the most problematic deficits is the lack of knowledge on the exact mechanism by which patients acquire pathogens in the healthcare environment. Transmission to, from and between patients, staff and the environment has not been fully elucidated and this means that it is impossible to prioritise or target infection prevention interventions for patient benefit. In order to focus on transmission dynamics, it was decided to embark on a study that might have the potential to pinpoint the exact pathway leading to healthcare-acquired infection (HAI). While an experimental unit would make it easier to track pathogens between reservoirs, this would not provide real-world data. So an intensive care unit (ICU) in a district general hospital (DGH) in Scotland was chosen for the surface-air-sampling study (SASS), which took place throughout 2015.

Elucidating dynamic transmission requires an indicator pathogen. *Staphylococcus aureus* represents a useful marker of hospital hygiene since it colonises 1 in 3 people, including staff, patients and visitors, and is thus found in air and on hands and surfaces, including equipment [1]. This organism was the obvious choice to investigate transmission between all major reservoirs in a clinical unit. It is also amenable to whole-cell genomic typing strategies.

This commentary summarises the study; why it was performed, how it was performed, and the main results, which were published in a sequence of three papers by the authors of this article [2,3,4]. The findings suggest the main direction of travel for the study pathogen, *S. aureus*, as well as highlighting the most important reservoirs in an ICU. It draws together the implications from all three papers and offers a series of recommendations for healthcare workers charged with controlling infection in the critical care environment.

## 2. Methods

### 2.1. Setting

The study took place in a 10-bed adult ICU in a semirural health board in Scotland [2,3]. The ICU has nine beds in the main area of the unit and one isolation room (Figure 1). At least 600 patients are admitted each year with a daily turnover of at least 1–5 patients. The case mix includes pneumonia, major trauma, sepsis, cardiac conditions and postoperative support. The ICU is mechanically ventilated with rates maintained at 10 air changes per hour at constant temperature and humidity. Detergent cleaning of general surfaces is performed daily by domestic staff, with near-patient sites cleaned twice daily by nurses using detergent wipes. Detergent wipes are also used to clean clinical equipment, with a once weekly bleach (Actichlor Plus™) disinfection [2]. Patients colonised or infected with hospital pathogens (e.g., methicillin-resistant *S. aureus* (MRSA); vancomycin-resistant enterococci; or *Clostridium difficile*) are isolated and the room or bed space disinfected with bleach on a daily basis.

### 2.2. Environmental Screening

The study ran for ten months, with a different morning chosen for sampling each month and with different intervals between study days [2,3]. Clinical staff were not told when screening was planned in order to circumvent any Hawthorne-type effect. Consideration was given to staffing levels and ICU bed occupancy (≥50%) before beginning the sampling protocol. There was a period of at least two hours after routine cleaning before systematic sampling began.

Double-sided dipslides coated with nutrient and staphylococcal-selective (Baird–Parker) agars were employed for hard surface sampling (Hygiena International, Watford, UK) [5]. Each slide was pressed firmly onto five near-patient sites for 10 s at a pressure of 25 g/cm^2^. Chosen sites were panels on the intravenous fluid pump and cardiac monitor; right and left bedrails and bed table [2,3]. The slides were incubated at 35 °C in CO_2_ for 48–72 h, depending upon colony size. Aerobic colony counts (ACC) per cm^2^ were categorised as follows: no growth; scanty growth <2.5 cfu/cm^2^; light growth 2.5–12 cfu/cm^2^; moderate growth 12–40 cfu/cm^2^; and heavy growth >40 cfu/cm^2^. Potential coagulase-positive staphylococci were captured from the selective agar, recultured and tested for coagulase production and antibiotic susceptibilities in accordance with routine laboratory protocol [2,3,4,5].

Settle plates (9 cm) were used for passive air sampling. These contained the same nutrient and selective agars in order to measure total counts and *S. aureus* (cfu/9 cm plate/h) [6]. They were placed on 1 metre high trolleys for one hour, with one in the isolation room and three in the main ICU (Figure 1). We performed automated air sampling at the same positions as the settle plate trolleys with an MAS-100 slit sampler based on the Andersen impactor principle (Merk, Germany) [7]. ICU air stream was directed onto nutrient and selective agar plates for 10 × 1 min [3]. The plates were processed using the same protocols for total bioburden and presence of *S. aureus* [2,3,4,5].

### 2.3. Patients, Visitors and Staff

Patients admitted to ICU are routinely screened for *S. aureus*; screening is repeated twice weekly and on discharge. Regular sampling of nose, perineum, wounds and urine enabled us to establish carrier patients (transient or permanent), staphylococcal colonization pressure, duration of staphylococcal shedding and investigate acquisition incidents occurring in ICU. Visitors were not screened, but staff voluntarily placed finger tips onto blood agar for enumeration and isolation of any possible *S. aureus*. Patients with confirmed *S. aureus* infection occurring >48 h after admission were assumed to be ICU-acquired and these were defined according to national guidelines [8].

### 2.4. Staphylococcal Genotyping

Staphylococcal isolates were sent to the Staphylococcal Reference Laboratory (National Infection Service, Public Health England, Colindale) for *spa* typing and MLST-CC assignments. Use of the *spa* server (http://spa.ridom.de/mlst.shtml), MLST database (http://saureus.mlst.net) and in-house PHE database helped to establish identity between strains [9]. Isolates with epidemiological links and related *spa* types were subjected to whole genomic sequencing (WGS) as previously described [10]. Single-nucleotide polymorphism (SNP) analysis was used to determine phylogenetic relationships between isolates at the core genome level (https://github.com/phe-bioinformatics/PHEnix). Any pairs or clusters of isolates with <50 SNPs between them suggested identity and these were explored further [11].

## 3. Results

Initial examination compared the surface bioburden of hand-touch sites, in order to model the level of microbial soil against the number of times the site was actually handled [2]. Staff were observed touching study sites from an average of 6/h (cardiac monitor) to 37/h (bed table) (Table 1) [2]. Just ten *S. aureus* (including one methicillin-resistant *S. aureus*: MRSA) were recovered from 500 screened sites around the patients’ beds. These comprised four from the left bedrail, two each from bed table and intravenous pump, and one isolate each from right bedrail and cardiac monitor. Seven isolates were linked with gross contamination (>12 cfu/cm^2^) of a specific site (*p* = 0.005), and six of these were recovered from the highest number of touched sites (bed table and bedrails) (Table 1).

While the bed table was the most frequently touched site, it did not deliver the highest amount of bioburden as expected (Figure 2). We wondered whether staff using alcohol gel from bottles on the bed table may have transferred gel to table surfaces, or generated microaerosol that settled on sampling sites. This premise was tested by removing the bottle of alcohol gel from one bed table in the middle of the ICU and rescreening the table during ten unannounced visits. Five of ten dipslides yielded >12 cfu/cm^2^, which was higher than the proportion from either bedrail and allowed us to confirm the relationship between touch frequency and surface soil (Figure 2). There is clearly a quantitative association between the number of times a site is touched and the amount of aerobic soil recovered from that surface [2]. Furthermore, there is a higher chance of isolating *S. aureus* from a surface if it is already heavily contaminated with microbial soil.

The second analysis involved modelling quantitative surface bioburden against microbial counts gathered from both passive and active air sampling within the 2 h sampling period on ICU [3]. Five hundred near-patient sites yielded quantifiable bioburden, ranging from no growth to heavy growth (>40 cfu/cm^2^) (Table 1). The microbiological surface standard chosen for these sites was <2.5 cfu/cm^2^, which gave an overall 47% failure rate [12]. Comparing the data from both air sampling methods allowed a proportionate comparison of pass or fail according to the surface standard.

Passive air sampling delivered values of 0–40 cfu/plate/h, with >2 cfu/plate/h recovered from 20 of the 40 plates (Table 2). This suggested a failure rate of 50%, if using the index of microbial air contamination (IMA) [6]. The IMA proposes a standard of ≤2 cfu/plate/h, while the standard for active air sampling is <10 cfu/m^3^ [7]. Fifteen of forty samples produced >10 cfu/m^3^, thus providing a failure rate of 37.5%. Thus, proportionate fails rate from passive air sampling (50%) was closer to the surface failure rate (47%) than the active air failure rate (37.5%) for ten study days.

The pass/failure rates from both air sampling methods were compared against the surface bioburden pass/fail rate on a site-by-site basis for each study day. There were just 19/40 (47.5%) pairs that agreed a pass or fail status between active air sampling data and surface bioburden, although settle plate data showed a closer relationship with surface counts (26/40: 65%) using the 2.5 cfu/cm^2^ benchmark [12]. Given that this benchmark has yet to become universally established, it was questioned whether pass/fail proportions for active air and settle plate counts would demonstrate a similar relationship with surface data if another surface standard was used. Consequently, all surface bioburden data was assigned pass or fail against a range of different standards from 1–20 cfu/cm^2^. The closest pass/fail agreement between any air parameter and specific surface standard occurred at 5 cfu/cm^2^ for settle plate data, with 70% agreement (Figure 3) [3].

We identified eleven patients with ICU-acquired staphylococcal infections occurring within the 72 h period encompassing each study day. Taking % bed occupancy into account, the number of these infections was plotted against total surface cfus per bed (all five bed sites) for Beds 2–10 for each individual study day. We ignored data from the isolation room because all patients with ICU-acquired infections were identified in the main ICU. Bed occupancy rate-adjusted ICU-acquired staphylococcal infection was associated with average surface count for patients in the main body of the ICU (*p* = 0.012) (Figure 4) [3].

The third analysis of data concerned results from whole genomic sequencing (WGS). WGS established 34 *S. aureus* clusters between reservoirs and patients, with another four pairs showing convincing phenotypical and epidemiological relationships (Table 3) [4]. There were 20 of 34 (59%) pairs that were highly related (<25 SNPs); these pairs linked a carriage strain with a strain causing acquired infection in the same patient, i.e., so-called autogenous or endogenous transmission [13]. Most of these infections were ventilator-associated (13 of 20), but there were also two central-line infections, four wound infections; one intra-abdominal infection and one abscess. The period of time between confirming colonisation and recovering the *S. aureus* causing infection was an average of 2.7 days (range 0–8 days). There were four other transmissions between four patient pairs with time intervals from 2–3 days to several months. Two of these pairs were highly related, but the relationship between the other two could not be verified because these isolates were EMRSA-15 and this strain was present elsewhere in the hospital [11]. Two patients had been on the same ward at the same time, but the isolation of their MRSA strains occurred over a 5 month period. The second MRSA pair involved two patients on ICU with just 4 days separating the isolation of their strains. At this time, these two patients were the only ones with MRSA in the ICU.

Four pairs of *S. aureus* linked hand-touch sites and patients, and all of these involved sites within the patients’ own, or adjoining, bed spaces (Table 3). One pair was classified as “uncertain” according to genomic identity definitions [14]. There were two linked isolates from bed table and cardiac monitor in adjoining bed spaces and another pair recovered from a bed table and bedrail on the same day, three bed spaces apart. There were five transmission episodes involving bedrails, with four of these implicating the left bedrail. Staff usually touch the bedrail on the patient’s right, and visitors more usually touch the bedrail on the left [2]. There were two pairs of strains involving the bed table, and the intravenous pump and cardiac monitor were also linked in two separate transmission episodes.

Despite 10 air changes/hour, there were four airborne *S. aureus* recovered and three of these were implicated in transmission links between a near-patient site and staff hands. There were no transmission episodes involving patients and staff hands, which was surprising because we always assume that HAI is associated with contaminated staff hands.

## 4. Discussion

This article describes a detailed study in one ICU in an attempt to identify transmission pathways between patients, staff and the environment using *S. aureus* (Figure 5) [2,3,4]. We found a relationship between the amount of times a site was touched and the total burden of microbial soil at that site. There was also a relationship between the amount of microbial soil on surfaces and the number of microbes in surrounding air, although the best relationship came from passive, rather than active, air sampling.

Air samples denote only a proportion of total surface bioburden, because microbial soil found on surfaces represents a combination of air deposition and direct contact. Thus, settle plates might be more useful as a routine monitoring strategy rather than a method for investigating outbreaks. Air sampling in isolation cannot detect surface contamination from other sources, such as handling, indirect contact and spillages [3].

WGS revealed that autogenous transmission was the most important direction of transmission, i.e., between patients’ colonised and infected sites. This is, perhaps, predictable, but it is reassuring to know that exogenous infections might be prevented by cleaning and disinfection, given that the next most common pathway occurred between patients and hand-touch sites around the bed [1]. We could not find any evidence for direct transmission between patients and the air, or between staff hands and patients, although *S. aureus* was recovered from both air and hands during the study.

Air samples were collected in the morning, which illustrates a major limitation of the study because airborne bioburden fluctuates significantly throughout the day and could yield values that are higher than found in this study. There are additional limitations, such as the fact that the study was performed in a single ICU only; there were just 10 sampling days in 10 months; patient demographics were not reported; there were no tangible data on the effectiveness of environmental cleaning; or whether patients were appropriately isolated, including compliance with contact precautions. It is also the case that staff and visitors were not screened.

We would, however, like to suggest a number of recommendations for preventing staphylococcal infection in ICU patients:

Firstly, frequently touched near-patient sites would benefit from targeted cleaning in the ICU, although the definitions of “cleaning” and frequency are not yet established; furthermore, there is contention over choice of cleaning fluids, i.e., whether we should use disinfectant or detergent or both [15]. There is increasing awareness that routine removal of microbial soil using the “one wipe; one site; one direction” principle is sufficient for infection prevention, rather than an attempt to kill all surface flora with disinfectants.

Secondly, visitor hand hygiene should be considered in an overall infection prevention strategy for ICU [16]. This is because visitors might have played a role in transmitting *S. aureus* in ICU since WGS identified numerous strains without any matches from sampled reservoirs. The risk from visitors’ hands remains an unexplored issue throughout hospitals in general.

Thirdly, given that patients are more at risk from their own staphylococcal strains, they should be screened routinely for *S. aureus* carriage on admission to ICU and regularly thereafter [13]. There are several effective decontamination packages for *S. aureus* carriage which can be employed following risk assessment.

Fourthly, the relationship between bioburden on hand-touch surfaces and in the air lends itself to further research; if routine settle plates can predict increased risk for infection, then this might be a valuable asset for infection prevention practitioners and much easier to perform than surface sampling, culture and interpretation [3].

## 5. Conclusions

This study collected *S. aureus* from staff hands, patients and environmental sites in one ICU and demonstrated potential transmission pathways between all reservoirs using genomic typing strategies. New strains are constantly introduced into critical care from colonised patients, which not only pose a major risk for the carrier but also to other patients through contamination of bed space sites. We could not demonstrate transmission involving staff hands or airborne staphylococci despite obvious presence. While one study cannot comprehensively define transmission hierarchies, the data clearly supports regular cleaning of near-patient hand-touch sites, patient screening and continued emphasis toward hand hygiene for everyone.

## Figures and Tables

**Figure 1 ijerph-17-02109-f001:**
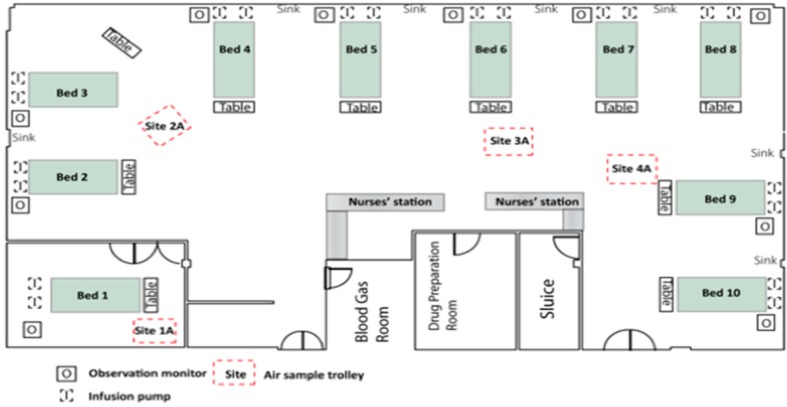
Aerial view of ICU (intensive care unit).

**Figure 2 ijerph-17-02109-f002:**
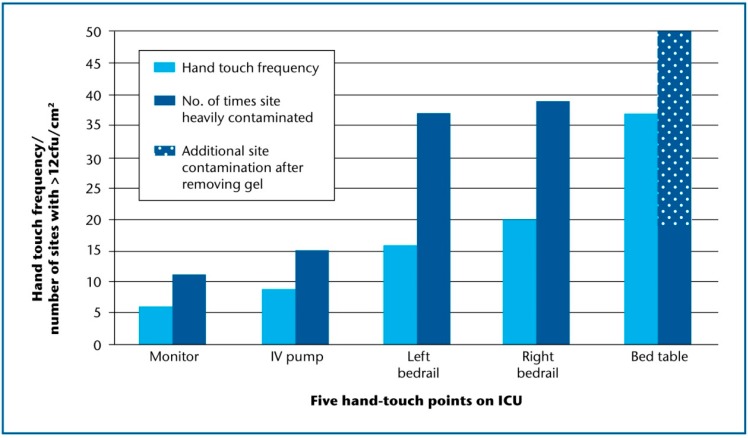
Hand-touch frequency and gross microbial soil for five near-patient sites on ICU. Average hand-touch frequency/site/h following ten observational audits; each site (n = 5) in ten bed spaces was screened on ten occasions; gross microbial soil defined as no. of screens exceeding 12 cfu/cm^2^; ICU: intensive care unit.

**Figure 3 ijerph-17-02109-f003:**
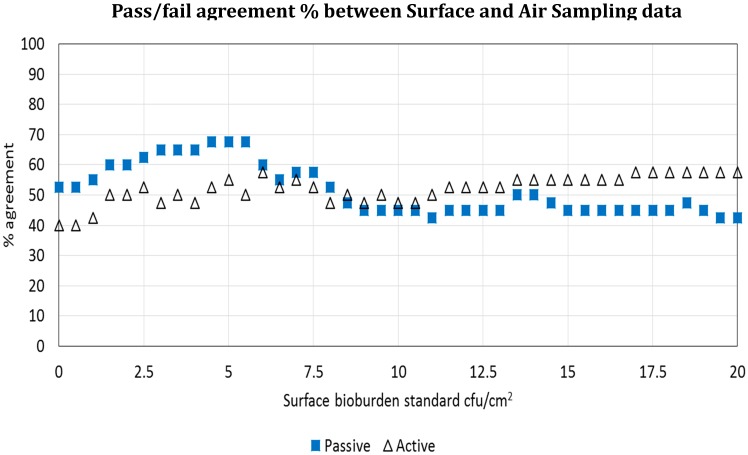
Is surface bioburden associated with air bioburden? Agreement between active and passive air sampling and surface bioburden using a range of surface standards from 0–20 cfu/cm^2^. The X axis shows the percentage pass or fail agreement between active and passive air data for each bioburden standard; the Y axis shows the surface bioburden value in cfu/cm^2^.

**Figure 4 ijerph-17-02109-f004:**
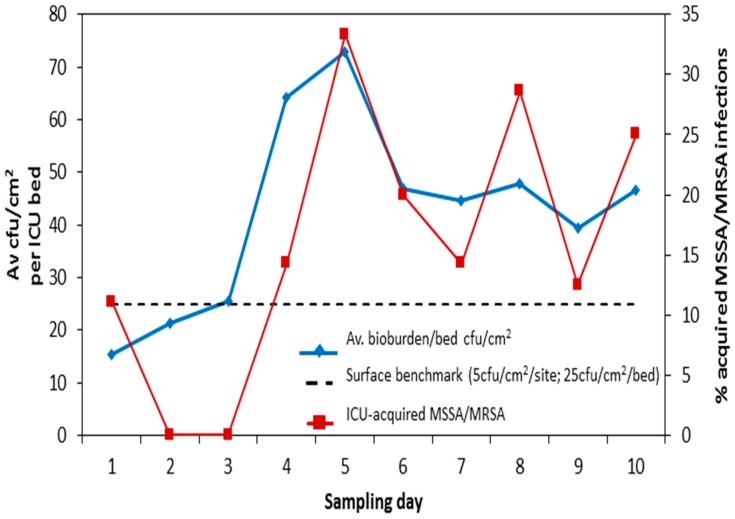
Is surface bioburden associated with clinical risk? Total bioburden (5 sites)/bed (cfu/cm^2^) plotted against % ICU-acquired *S. aureus* infection (adjusted for bed occupancy) for Beds 2–10 on 10 sampling days. ICU: intensive care unit; MSSA: methicillin-susceptible *S. aureus*; MRSA: methicillin-resistant *S. aureus*.

**Figure 5 ijerph-17-02109-f005:**
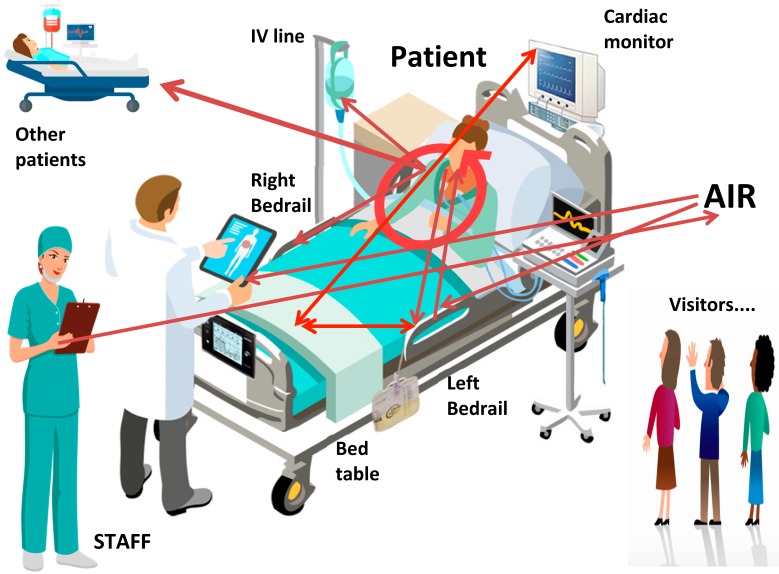
Transmission pathways of *S. aureus* on one ICU. ICU: intensive care unit; MSSA: methicillin-susceptible *S. aureus*; MRSA: methicillin-resistant *S. aureus*.

**Table 1 ijerph-17-02109-t001:** Microbial soil categories for five hand-touch sites on ICU.

Site	No Growth	Scanty Growth < 2.5 cfu/cm^2^	Light Growth > 2.5–12 cfu/cm^2^	Moderate Growth > 12–40 cfu/cm^2^	Heavy Growth > 40 cfu/cm^2^	No. of Hygiene Fails (>2.5 cfu/cm^2^)
Infusion Pump	16	47 **MSSA**	22	13 **MSSA**	2	37/100: 37%
Cardiac Monitor	45	28	16 **MSSA**	9	2	27/100: 27%
Right Bedrail	6	38	17	27	12 **MSSA**	56/100: 56%
Over-bed Table	13	35	33 **MSSA**	16 **MSSA**	3	52/100: 52%
Left Bedrail	6	31	26	25 **MSSA × 2**	12 **MSSA & MRSA**	63/100: 63%

MSSA: methicillin-susceptible *S. aureus*; MRSA: methicillin-resistant *S. aureus*; hygiene standard for surfaces: <2.5 cfu/cm^2^ (ref. [6]); average surface fail = 47% (range: 27%–63%).

**Table 2 ijerph-17-02109-t002:** Microbial burden categories for air (active and passive sampling) and hygiene fails according to standards.

**Passive Air Sampling** **n = 40**	**No Growth**	**Scanty Growth** **0–2 cfu/plate**	**Light Growth** **> 2–10 cfu/plate**	**Moderate Growth** **> 10–40 cfu/plate**	**Heavy Growth** **> 40 cfu/plate**	**No. of Hygiene Fails** **> 2 cfu/plate/h**
Air settle cfu/plate/h	1	19 **MSSA**	18	2	0	20/40 = 50%
**Active Air Sampling** **n = 40**	**No Growth**	**Scanty Growth** **0–2 cfu/m^3^**	**L. Growth** **> 2–10 cfu/m^3^**	**Mod. Growth** **> 10–40 cfu/m^3^**	**Heavy Growth** **> 40 cfu/m^3^**	**No. of Hygiene Fails** **> 10 cfu/m^3^**
Air sampler cfu/m^3^	1	6	18 **MSSA × 2**	15 **MSSA**	0	15/40 = 37.5%

MSSA: methicillin-susceptible *S. aureus*; MRSA: methicillin-resistant *S. aureus*; hygiene standard for air (passive): ≤2 cfu/9 cm^2^ plate/h (ref. [6]); hygiene standard for air (active): <10 cfu/m^3^ (ref. [7]); overall, 50% passive air samples fail standards; 37.5% active air samples fail standards.

**Table 3 ijerph-17-02109-t003:** Whole-genome sequencing (WGS) categories and pathways, lineage, sites, intervals (days) and SNP (single-nucleotide polymorphism) differences of *S. aureus* clusters in a ten-bed ICU during a ten month study.

WGS Category	Transmission Pathway	Lineage (MLST-CC)	Patients and Sites Involved	Days between Clusters	No. SNP Differences
**Highly likely [10]**	1. Autogenous	8	Nose & Resp	2	<5
	2. Pt ↔ fomite (touch site)	5	Pt. 2 Resp, bed 3 → IVP, bed 3	3	<5
	3. Pt ↔ fomite (touch site)	5	Pt. 2 Resp, bed 3 ↔ R/Rail, bed 3	3	<5
	4. Autogenous	15	Nose & Resp	5	<5
	5. Autogenous	15	Nose ↔ CLT	5	<25
	6. Autogenous	22 (MRSA)	Pt. 4 Per & Pt. 4 DRF	2	<5
	7. Autogenous	22 (MRSA)	Nose & Resp	2	0
	8. Autogenous	22	Nose & Resp	1	<5
	9. Pt ↔ fomite (touch site)	22 (MRSA)	L/Rail ↔ Pt. 4 Per & Pt. 4 DRF	1	<5
	10. Autogenous	30	Resp & Nose	4	<5
	11. Autogenous	30	Nose & Resp	2	<5
	12. Autogenous	30	Pt. 7 Nose & Pt. 7 Per/Wound	5	<5
	13. Autogenous	30	Nose & Wound	1	<5
	14. Autogenous	45	Nose ↔ Resp	1	<25
	15. Autogenous	45	Nose ↔ Resp	2	<5
	16. Autogenous	45	Resp ↔ Nose	2	<25
	17. Autogenous	45	Pt. 3 Per ↔ Pt. 3 Wound	3	<5
	18. Air ↔ fomite	45	Air, beds 5–7 ↔ L/Rail, bed 7	0	<5
	19. Fomite ↔ fomite	45	Table ↔ CM	0	0
	20. Autogenous	7	Pt. 6 nose ↔ Pt. 6 CLT	8	<10
	21. Autogenous	34	Nose ↔ Resp ↔ Thr	2	<25
	22. Autogenous	59	Nose ↔ Resp	5	<25
	23. Autogenous	59	Nose ↔ Resp	0	<25
	24. Autogenous	188	Resp ↔ Nose	0	<10
	25. Autogenous	121	Abscess ↔ Nose	2	<10
	26.Staff hand ↔ air	25	Hand ↔ Air, beds 5–7	43	<5
	27. Staff hand ↔ air	25	Hand ↔ Air, beds 8–10	43	<5
**Possible**	28. Pt ↔ Pt Cross-infection	59	Wound ↔ Nose & Resp	2	<25
	29. Pt ↔ Pt Cross-infection	1	Nose ↔ Nose	4	<25
**Uncertain [14]**	30. Pt ↔ fomite (touch site)	5	Resp, bed 2 ↔ L/Rail, bed 2	4	<50
	31. Staff hand ↔ air	5	Hand ↔ Settle plate	50	<25
	32. Pt ↔ Pt Cross-infection	22 (MRSA)	Per ↔ Nose	161	<25
	33. Pt ↔ Pt Cross-infection	22 (MRSA)	Nose ↔ Nose	3	<25
	34. Fomite ↔ fomite	30	L/Rail, bed 4 ↔ Table, bed 7	0	<25
***Presumed*** (Phenotypic and epidemiologic relationships only)	1. Autogenous *	30	Pt. 5 Nose → Pt. 5 Resp Matching antibiograms	1	N/A
	2. Autogenous *	45	Pt. 8 Nose→ Pt. 8 Wound Matching antibiograms	4	N/A
	3. Autogenous *	1	Nose → Wound Matching antibiograms	0	N/A
	4. Pt ↔ Pt Cross-infection *#	7	Pt. 6 Nose/CLT → Pt. 9 Resp	48	N/A

**Key**: ICU = Intensive Care Unit; Pt = patient; Resp = respiratory secretions; DRF = drain fluid; IVP = intravenous pump; CTL = central line site; Per = perineum; L/R Rail = left/right bedrail; CM = cardiac monitor; Thr = throat; N/A = unavailable for *spa* typing or WGS. * Matching antibiograms included MICs performed using VITEK2™; # these patients were allocated the same bed space on ICU 3 weeks apart. All *S. aureus* from Pt 6 (including sputum) had matching antibiograms, which were unique within the study. Pt 6 stayed in ICU for 25 days. Pt 9 had eczema and carried an unrelated nasal *S. aureus*.

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
