# Peer review of "Dynamic Transmission of Staphylococcus Aureus in the Intensive Care Unit"

_ijerph, 2020, doi:10.3390/ijerph17062109_

Round 1

Reviewer 1 Report

Title: Dynamic Transmission of Staphylococcus aureus in the Intensive Care Unit

First author: Claire E. Adams

Journal: International Journal of Environmental Research and Public Health

Overall decision of the manuscript

The recommendation is to accept the article after minor revisions.

Comments to the author

In the article, the authors summarize the findings of three of their own published articles. The articles are all studies belonging to the Surface Air Sampling Study (SASS), which aims to determine pathways leading to healthcare acquired infections. Staphylococcus aureus was used as a model organism. The studies looked at the relationship between touch frequency and contamination of surfaces, modeling quantitative surface bioburden against microbial counts, and WGS to determine transmission.

They describe that in the first study, they found a relation between touch frequency and contamination of surfaces. In the second study, they describe that 37-50% of all surfaces has a higher bioburden that the standard of 2.5 cfu/cm2. Furthermore, they combine the results of active and passive air sampling with surface sampling, to determine the method with the highest agreement. In the final article, they looked at the WGS data to determine clusters of S. aureus between patients, the environment and healthcare workers. They identified mainly autogenous transmission. No transmission between health care workers and patients were identified. However, the messages as can be deduced from the data are extremely important , as, although not the majority, there is still a part of exogenous infections that can be prevented. However the authors interpret their results a bit in a pessimistic way. In the discussion

No limitations or strengths of the study are given, as these were discussed in the original publications?

They suggest four recommendations for preventing staphylococcal infections in ICU patients.

Major comments

I have no major comments about the data presented in the article. However, no abstract was included in the PDF file and the figures were also missing from my upload. The figures can be looked up in the original articles, but it makes it difficult to review them. Furthermore, it is not mentioned clearly in the article that the authors were also authors from the studies described in the article. This is only indirectly stated (for example by the use of “we” in line 111) or from looking at the author list of the original articles. I feel like it is important information for the readers to know this.

Minor comments

40           The references of the original articles are missing.

48           Throughout the article, they use the abbreviation ICU. Here the abbreviation ITU is used, which is confusing.

51           mechanical ventilation rate of 10; ventilation rate or circulation rate ? rooms at under- or overpressure? Or neutral?

51         routine cleaning; daily?

55           What do you mean with hospital pathogens? This can be interpreted in many different ways, depending on the reader.

56           Is the room cleaned with bleach after isolating the patient or during the stay of the patient?

Please be clear on using words such as (routine) cleaning and disinfection; terms are used differently over countries; cleaning is sometimes interpreted as water/soap or dry and bleach e.g is disinfection. Give definitions you use in this appear.

58           For me, it’s not clear what the exact sampling moments were.

59           staff; which staff? Cleaning staff?

63;          nutrient? Which nutrient? Staphylococcal selective; which agar?

65           pressure 25g/cm2 how did you measure or perform this?

67           48 to 72hours? Is it 48 or 72 hrs?

67           sample sites; was it by suing a fixed cm2 surface to be sampled?

75           during activities? Measured?

87           assumptions hospital (ICU) acquired = after 48 hrs admissions; these assumptions are for this study rough assumptions, especially for the ICU; were patients already admitted before ICU admission? Is the 48 hrs on the ICU or in the hospital? ; were all isolates sent for typing; then why should you use this low sensitive definition? Just rely on typing results

91           including environmental strains?

104        This information (about the touch frequency) is not included in Table 1. However, it is describe in Table 1 from Adams et al. 2017. Examining the association between surface bioburden and frequently touched sites in intensive care. The table in the current article is Table 1 from Smith et al. 2018 Is there an association between airborne and surface microbes in the critical care environment? Also, the methods of observation is missing in this paper

108        When someone states "another seven" I assume that these isolates are different isolates than the seven described before. However, 6 out of the "another" 7 isolates are also part of the before described seven isolates. It is better if this is rephrased.

111         before assuming leakage or use of hand alcohol on the bed table, maybe the material of the bed table is less at risk for contamination? Removing hand alcohol can also increase hand contamination and with that increase bed table contamination.. this result is a bit too much assumptions. Furthermore, as this study is on S aureus, these results highly depends on the patients in the room; are these S aureus carriers or not.

145         This paragraph feels more like a discussion section than a result section.

150         The number of screened patients/staff is not mentioned, only the number of patients with an acquired infection is mentioned.

160        I understand why they are phrasing it this way, but technically it is not true. Of the 34 possible transmissions, 33 have a SNP difference of <25 SNPs. Please use definitions in the method section as it seems that the cutoff you use is 25 SNP (n=320, but definition for a cluster is different (n=34)? Autogenous is a difficult term for m; exogenous or endogenous? I assume you mean endogenous; and that the 20 pairs were strains from colonization site and infection site from the same patient.? Please clarify this in your paper. Actually, these infections are not so interesting for the purpose of this paper. It just reassure the majority of S aureus infection s are endogenous, but the exogenous ones can be prevented by cleaning and disinfection; please make this clear

186         may be due to high compliance of hand hygiene?

190         why using the word “hoped” it looks like the authors are disappointed in their own results; I am not; there is still a vast amount of transmission from the environment to patients; the fact that endogenous or autogenous infection “(please define “infection..) was already known and is important to reconfirm; other preventive measures should be used to prevent these infections.

199         “occasionally found”; be more precise; even low detection numbers can be important as these measure (air) depend on the number of measurement done while a carrier or S aureus infected patient was in the room; please correct for this or make clear.

209         were visitor HH included in the observations?

211         The sentence should be rephrased. Now it implicates that the risk from visitors’ hands remains and unexplored issue throughout the hospital because visitors might have played a role in transmitting S. aureus.

217         This is not a recommendation for preventing staphylococcal infections, but more a recommendation for future research.

226         “nor any evidence”this is too hard… you find S aureus in the air; this is important, especially in the light if prevention measures (an isolation room? Negative pressure? 0 etc. now it looks that they conclude that the study was not worthwhile and had negative outcome, whereas I am more enthusiastic.. not showing a relation today, will not conclude that these strains in the air were form no value..but in line 229 the authors ends with a more optimistic end. Please rephrase

229         In line 210 they state that visitors’ hands remains an unexplored issue and in line 226 they state that no transmission between staff and patients was demonstrated. The conclusion and continued emphasis toward hand hygiene for staff and visitors does not follow these statements. No information is given about hand hygiene compliance, which could indicate good hand hygiene and the reason to continue to emphasis this. Furthermore, there is no mention about current emphasis to visitor hand hygiene. If this is not yet done, it also cannot be continued.

252         If this is indeed figure 1 from Adams et al. 2017, the legend of the figure is missing in the description.

268         This figure was the hardest to find in the original articles. I believe that it is figure 1 from Dancer et al. 2019 Tracking Staphylococcus aureus in the intensive care unit using whole-genome sequencing. If that is true, I believe the description should be more elaborate to make it easier to understand. Furthermore, this figure is only introduced in the discussion, which is not usual to do. I am not sure that it is necessary for the article to include the figure.

Reviewer 2 Report

It's an interesting and well-written paper except for some minor revision as following,

1-A little bit tedious for Table results. The tables are supposed to be condensed to lighten the paper.

2- Quality of Figures are not readable and should be improved for readability.

Reviewer 3 Report

The topic of the study is especially timely now and always it is important to control the transmission of the diseases in health care units. The work was limited to one unit, but however, it gives a good view of the hygiene conditions in ICUs in general. It is important to find out hot spots where the microbial contamination is the most important to control. In general the manuscript is well and clearly written. E.g. in page 2 there were acronym ITU, instead of ICU. There are some other misspellings and need for clarification to get the text unambiguous. The methodology used is sufficient. The description of the management of ICU is described. Are there any routine quality control of measures used to check of cleanliness after cleaning and disinfection. The sampling sites were selected to represent the most probable sites to contaminate due touching and also samples from air were taken to find out transmission of airborne bacteria. The quality of the figure 1 was poor (at least in my version) that need to revise. In the results chapter page 5, and also in the Figure 2, you mention touching frequency. What do you mean with frequency. Touch per hour, or touch per min? or touch per working shift. Please clarify. The analysis of bioburden is based on passive deposition plates. The method is sensitive for air flow velocity and direction in the vicinity of the sampling site. Could you please inform the average air velocity, or at least discuss it, in the rooms. In isolation rooms the air exchange rate is quite high giving much higher velocities than in ordinary patient room. In figure 3 you have named the axes % agreement. Could you make more specific one (so percentage or ratio of what). The analysis of the genomic type of S. aureus was useful to get hint of the origin of the microbial contamination. That is important to know when preventative actions are designed. In discussion chapter you are presenting recommendations. The data of the study is quite limited for generalization of the results. However, it is a good way to conclude for practitioners how to limit microbial contamination in ICU environment. The conclusion chapter is included in the discussion chapter which could be also under as its own title. The style of the journal can decide this matter.

Author Response

This manuscript is a resubmission of an earlier submission. The following is a list of the peer review reports and author responses from that submission.